# OpenReview forum: "Fantastic Reasoning Behaviors and Where to Find Them: Unsupervised Discovery of the Reasoning Process"
_ICLR.cc/2026/Conference — Submitted to ICLR 2026_

### Official Review · Reviewer_CgVw · 2025-10-31

**Soundness:** 3
**Presentation:** 3
**Contribution:** 2
**Rating:** 4
**Confidence:** 3

**Summary:**

This paper proposes an unsupervised framework for explaining the reasoning behavior of LLMs: Reasoning behavior Interpretability via Sparse auto-encoder (RISE). This framework explains the reasoning behavior by training sparse auto-encoders to distinguish inference vectors. Experiments show that reflection and backtracking unpacking functions occupy separable regions in the decoder column space. The paper also analyzes the impact of targeted interventions, response length, and other factors on the model's reasoning behavior.

**Strengths:**

1. This paper proposes an unsupervised framework for interpreting reasoning behavior, which requires no human intervention and can intuitively reflect the inference vectors during model reasoning.

2. This paper investigates the model's reasoning behavior through ample experiments. The results suggest that reflection and backtracking unraveling functions occupy separable regions in the decoder column space. Furthermore, the paper analyzes several factors influencing the model's reasoning behavior.

**Weaknesses:**

1. Limited Domain Focus: The experimental section of the paper only used the Mathematics dataset (MATH) to analyze mathematical reasoning behavior. Adding reasoning tasks from other domains, such as common-sense reasoning and logical reasoning, might strengthen the experiment's persuasiveness.

2. In the experiments in Section 4.3, regarding the experimental results in Figure 3, the paper explains the statement that "the comparison between reflection and backtracking shows only moderate differences, while the separation of 'other' behaviors from reflection or backtracking is significantly stronger" as "reflection and backtracking occupy more overlapping representational subspaces." Are there any experimental results to support this conclusion?

**Questions:**

Please see the Weaknesses.

---

> ### Author Response · Authors · 2025-11-25
> **Responses to Reviewer CgVw**
>
> We thank Reviewer CgVw for taking the time to review our work. We’re pleased that the reviewer found our claims to be supported by “ample experiments” and appreciate the thoughtful and constructive suggestions provided. To address the remaining concerns, we offer detailed point-by-point responses below.
>
> **[Q1: Domain Generalization]**: Thanks for the question. To further evaluate the generality of our approach, we performed additional experiments involving positive and negative steering using the three behavior vectors: Reflection, Backtracking (Section 4.4; Results in Table R1), and Confidence (Section 5.1; Results in Table R2). We extended our evaluation to two new datasets outside the math domain, i.e., GPQA-Diamond and KnowLogic, which focus on commonsense and logical reasoning, respectively.
> The results, presented in Table R1/R2, show that applying these steering vectors leads to predictable changes in behavior counts in both positive and negative directions, demonstrating the ability of the vectors to control reasoning modes. This provides further empirical support for the robustness and generalizability of our framework across diverse reasoning domains.
>
> Details: In Tables R3 and R4, the numbers indicate the count of corresponding steps under positive, negative, or no steering (vanilla).
> Table R1. Causal effects of Reflection/Backtracking vectors.
>
> |     Steering Method | GPQA Reflection | GPQA Backtracking | KnowLogic Reflection | KnowLogic Backtracking |
> |---------------|------------------|--------------------|-----------------------|-------------------------|
> | **Vanilla**   | 53.23            | 11.83              | 35.56                 | 5.42                    |
> | **Positive**  | 62.77            | 20.31              | 51.00                 | 9.38                    |
> | **Negative**  | 45.42            | 6.47               | 25.99                 | 2.33                    |
>
> Table R2: Causal effects of confidence vectors.
>
> |     Steering Method   | GPQA Reflection | GPQA Backtracking | KnowLogic Reflection | KnowLogic Backtracking |
> |---------------|------------------|--------------------|-----------------------|-------------------------|
> | **Vanilla**   | 53.23            | 11.83              | 35.56                 | 5.42                    |
> | **Positive**  | 61.37            | 16.18              | 48.16                 | 5.98                    |
> | **Negative**  | 37.25            | 6.91               | 20.04                 | 3.51                    |
>
>
> **[Q2: Difference between reflection/transition and others]**: Thanks. We’d like to clarify that the silhouette scores in Figure 3 are supporting the claim that “the comparison between reflection and backtracking reveals only modest differences, whereas the separation of “other” behaviors from either reflection or backtracking is substantially stronger. This suggests that reflection and backtracking occupy more overlapping representational subspaces.”
>
> Specifically, Higher silhouette scores stands for less overlap and higher separability, as shown in Figure 3, both the curve of “others vs backtracking” and “others vs reflection is larger than the curve “reflection vs backtracking”, means that reflection and backtracking is more overlapped compared with others.

---

### Official Review · Reviewer_HCkk · 2025-11-01

**Soundness:** 4
**Presentation:** 3
**Contribution:** 3
**Rating:** 4
**Confidence:** 4

**Summary:**

The paper presents RISE, an unsupervised framework based on sparse auto-encoders (SAE) in large language models. It aims to identify reasoning vectors in the activation space that correspond to behaviors such as reflection and backtracking, showing that these occupy separable regions in the SAE decoder space. It further demonstrates causal intervention, enabling targeted modulation of reasoning behaviors and model confidence beyond explicit human supervision.

**Strengths:**

1. The paper is well-structured, logical, and clearly written, making the complex concepts easy to follow.
2. The core idea of using SAEs to analyze the reasoning patterns of LLMs is novel and interesting. The demonstration that manipulating these disentangled features leads to a clear and observable enhancement or suppression of the corresponding reasoning behaviors is a significant strength.
3. The experiment to identify "confident reasoning vectors" by optimizing for entropy in the SAE decoder space is compelling. The subsequent intervention, which demonstrably boosts the model's confidence.

**Weaknesses:**

- In Figure 2, the separation between the *Reflection* and *Backtracking* clusters does not appear very distinct, and the normalized Silhouette scores in Figure 3 peak at only around 0.6, which is close to the boundary of meaningful separation. It would be helpful to clarify whether the clustering in the decoder space can be considered sufficiently reliable under these conditions.
- In Section 4.5, the paper suggests that cluster formation in the mid-to-late layers reflects response length, but this relationship is not clearly visible from the figures. Including a quantitative analysis (such as the Figure 3) could better support the claimed link between cluster compactness and response length.
- Figures 6 and 8 indicate that the c*onfidence vectors* may influence reasoning in a way similar to r*eflection* and b*acktracking vectors*. It might be useful to analyze the similarity or correlation among these three groups to determine whether “confidence” represents a distinct factor or overlaps with the other two behaviors.
- The experiments are currently limited to a single model (DeepSeek-R1-Distill-Qwen-1.5B). Extending the analysis to other models, such as the base Qwen series or architectures like Llama, would help demonstrate the consistency and robustness of the findings.
- If the authors can address these concerns clearly in the rebuttal, I would be inclined to raise my overall rating.

**Questions:**

See weakness

---

> ### Author Response · Authors · 2025-11-25
> **Responses to Reviewer HCkk**
>
> We sincerely thank Reviewer HCkk for describing our work as “novel” and “interesting,” and for the positive recognition of the manuscript’s structure and writing. We also appreciate the constructive suggestions, which were highly valuable in improving the quality of our work. Below, we provide point-by-point responses to address each concern.
>
> **[Q1: Reliable of Clusters]**: Thanks for the question. We understand the concern that, although Figures 2 and 3 show visually separated clusters for reflection and backtracking, there may still be some degree of overlap between the two behaviors. To further verify the reliability of this separation, we conducted additional causal‑effect experiments on GPQA‑Diamond.
> Specifically, following the intervention procedure in Figure 4, we steer the model using the centroid of the reflection cluster in both positive and negative directions. We observe that the frequency of reflection changes in a consistent and monotonic manner: 53.23 → 62.77 → 45.42 for {vanilla → positive → negative}. In contrast, the frequency of backtracking remains nearly unchanged, 11.83 → 11.73 → 11.57 under the same interventions.
> These results demonstrate that even if the two clusters are not perfectly separable in the 2‑D visualization, the underlying SAE directions have distinct and behavior‑specific causal effects. This provides strong evidence that reflection and backtracking correspond to reliably different subspaces in the decoder geometry.
>
> **[Q2: Quantitative Analysis of Length Cluster]**: Thanks for the suggestion. We provide the quantitative analysis in Figure 9 of Appendix Section C. The results show that the early layers exhibit weak separation with respect to response length, whereas the mid-to-late layers demonstrate stronger alignment with the response length.
>
> **[Q3: Clarification of Confidence/Reflection/Backtracking Behaviors]**: Thanks for the question. We did analyze the correlation between confidence, reflection, and backtracking behaviors through causal intervention, as described around Line 465. Specifically, when we steer the confidence vector in a negative way, we observe that reflection steps drop from 90.53 to 33.77, and backtracking steps decrease from 35.50 to 5.93. These causal effects indicate that the confidence vector has a significant influence on the model’s reasoning behavior, impacting both reflection and backtracking.
>
> **[Q4: More Models]**: Good point. We would like to clarify that we have already included experimental results on the Qwen models. Specifically, Figure 5 presents causal intervention experiments conducted on Qwen3-8B, demonstrating the effectiveness of our framework on this model. We have revised the experimental descriptions in Line 225 to improve clarity.

---

### Official Review · Reviewer_qNtY · 2025-11-03

**Soundness:** 3
**Presentation:** 4
**Contribution:** 4
**Rating:** 4
**Confidence:** 3

**Summary:**

This paper proposes RISE, an unsupervised framework that trains sparse auto-encoders on sentence-level chain-of-thought activations to discover “reasoning vectors”, which are directions in activation space that encode interpretable reasoning behaviors.  The paper shows these vectors form coherent clusters, and that test-time interventions along the vectors can amplify or suppress behaviors, changing inference trajectories without retraining. They further observe a structural axis related to response length and identify confidence vectors by optimizing for low entropy, which shifts outputs toward more confident reasoning.

**Strengths:**

1. Unsupervised SAEs on step-level activations reveal interpretable reasoning vectors and enable causal behavior steering without retraining.

2. The method RISE is coherent with plausible identifiability arguments and consistent cross-benchmark evidence supporting the main claims.

3. The findings are valuable for interpretability and practical control of LLM reasoning.
The paper presents a clean end-to-end pipeline with intuitive figures and explanations that make replication straightforward.

**Weaknesses:**

1. The paper claims that the behavior labels rely on an LLM-as-judge, but does not demonstrate robustness to model choice or prompt wording.

2. The composition of multiple behavior vectors may introduce interference due to non-orthogonality.

3. The identifiability of RISE claim depends on sparsity and incoherence assumptions, which are not empirically validated on real activations in this paper.

4. The experiments are confined to math reasoning, raising my concerns about domain generality.

**Questions:**

1. Could the authors provide cross-annotations, e.g., multiple annotators on the same samples, to demonstrate consistency and robustness?

2. Could the authors demonstrate RISE’s performance in other domains, such as code generation, to test transferability?

---

> ### Author Response · Authors · 2025-11-25
> **Responses to Reviewer qNtY [Q1-Q3]**
>
> We appreciate that Reviewer qNtY recognized the value of our contributions and rated the presentation as excellent, as well as the findings as “valuable” to the community. To further address the reviewer’s concerns, we provide point-by-point responses below.
>
> **[Q1: Details about LLM-as-judge]**: Good suggestion. We have added a detailed description of the prompt used for LLM-based annotation in Section D of Appendix. Additionally, we conducted further experiments to evaluate the consistency across different annotation methods. Specifically, we used the same prompt from Section D to query three models, including GPT-5, GPT-4o, and Claude Sonnet 4.5, and compared their annotations with a keyword-matching baseline as a reference. The results, presented in Table R1, show strong overall consistency, with agreement ratio (Defined as the proportion of steps sharing the same annotation relative to all steps) exceeding 85% across all methods. Notably, GPT-4o and GPT-5 exhibit the highest agreement, reaching up to 94%. These results support the reliability and consistency of our annotation methodology.
>
> Table R1: Agreement ratio of different annotation methods.
> |                   |   Claude Sonnet 4.5 |   Keywords |   GPT-4o |   GPT-5 |
> |:------------------|--------------------:|-----------:|---------:|--------:|
> | Claude Sonnet 4.5 |                1.00 |       0.86 |     0.88 |    0.85 |
> | Keywords          |                0.86 |       1.00 |     0.86 |    0.85 |
> | GPT-4o            |                0.88 |       0.86 |     1.00 |    0.94 |
> | GPT-5             |                0.85 |       0.85 |     0.94 |    1.00 |
>
>
> **[Q2: Interference Effect of multiple Behaviors]**: Good point. We would like to clarify that in our study, we do not combine non-orthogonal behavior vectors. For the results shown in Figures 5 and 8, we select only the top relevant vectors corresponding to a single behavior type. As a result, there is no interference from other behavior vectors, and the findings are not affected by potential non-orthogonality across behaviors. Additionally, we acknowledge the potential challenges that may arise when combining multiple non-orthogonal behavior vectors to modify the model’s reasoning style. We agree that this raises an interesting research question: how can one steer reasoning to follow multiple diverse behaviors effectively? Exploring the potential benefits and mechanisms for such multi-behavior control is a promising direction, and we plan to investigate it in future work.
>
>
> **[Q3: Sparsity/Incoherence on Real Activation]**: Good catch! We conducted additional experiments to empirically validate the sparsity and incoherence assumptions. As shown in Table R2, since the ground-truth dictionary W in the theorem is not directly observable, we use the reconstructed activations from the sparse autoencoder for evaluation. Given the consistently low reconstruction loss in Table R2, this serves as a reliable proxy. We observe that the maximum cosine similarity among recovered dictionary vectors is clearly bounded by a valid value $\mu < 1$, and the sparsity level k (defined as the number of active dictionary vectors used to reconstruct each activation) is also well within the theoretical bound. These results provide empirical support for the assumptions underlying our theoretical analysis.
>
> Table R2, Results of Sparsity and Incoherence assumptions of R1-1.5B.
>
> | Layer | k-Sparsity | Loss           | Max Cosine Similarity  |
> |-------|--------------|----------------|---------|
> | 1     | 345          | 6.2e-4         | 0.2945  |
> | 2     | 129          | 1.1e-3         | 0.2877  |
> | 3     | 93           | 1.6e-3         | 0.2148  |
> | 4     | 49           | 1.8e-3         | 0.1610  |
> | 5     | 46           | 2.0e-3         | 0.2120  |
> | 6     | 46           | 2.0e-3         | 0.1812  |
> | 7     | 57           | 2.0e-3         | 0.1730  |
> | 8     | 35           | 2.5e-3         | 0.1554  |
> | 9     | 30           | 2.5e-3         | 0.1349  |
> | 10    | 25           | 3.1e-3         | 0.1203  |
> | 11    | 24           | 3.8e-3         | 0.1159  |
> | 12    | 22           | 3.9e-3         | 0.1239  |
> | 13    | 21           | 4.2e-3         | 0.1217  |
> | 14    | 19           | 5.0e-3         | 0.1011  |
> | 15    | 19           | 5.4e-3         | 0.1085  |
> | 16    | 16           | 6.2e-3         | 0.1004  |
> | 17    | 14           | 7.0e-3         | 0.1055  |
> | 18    | 13           | 8.3e-3         | 0.0963  |
> | 19    | 11           | 1.0e-2         | 0.0994  |
> | 20    | 10           | 1.2e-2         | 0.0987  |
> | 21    | 8            | 1.5e-2         | 0.0932  |
> | 22    | 7            | 2.0e-2         | 0.1041  |
> | 23    | 6            | 2.4e-2         | 0.0936  |
> | 24    | 5            | 2.9e-2         | 0.0954  |
> | 25    | 5            | 3.7e-2         | 0.0943  |
> | 26    | 5            | 5.1e-2         | 0.0933  |
> | 27    | 6            | 7.8e-2         | 0.0922  |
> | 28    | 23           | 7.8e-3         | 0.1219  |

---

> ### Author Response · Authors · 2025-11-25
> **Responses to Reviewer qNtY [Q4]**
>
> **[Q4: Domain Generality]**: Great suggestions. We conducted additional positive and negative steering experiments using the three identified vectors: the Reflection and Backtracking vectors from Section 4.4 (Results in Table R3), and the Confidence vector from Section 5.1 (Results in Table R4). To evaluate generalization beyond the math domain, we applied these interventions to two new benchmarks: GPQA-Diamond and KnowLogic[1], which assess commonsense and logical reasoning, respectively.
>
> As shown in Table R3/R4, the vectors extracted via SAE can effectively modulate reasoning behaviors, with the number of behavior-specific steps increasing or decreasing in accordance with the direction of steering. These results further validate the effectiveness and cross-domain generality of our framework.
>
> Details: In Tables R3 and R4, the numbers indicate the count of corresponding steps under positive, negative, or no steering (vanilla).
>
> Table R3: Causal effects of Reflection/Backtracking vectors.
>
> |     Steering Method | GPQA Reflection | GPQA Backtracking | KnowLogic Reflection | KnowLogic Backtracking |
> |---------------|------------------|--------------------|-----------------------|-------------------------|
> | **Vanilla**   | 53.23            | 11.83              | 35.56                 | 5.42                    |
> | **Positive**  | 62.77            | 20.31              | 51.00                 | 9.38                    |
> | **Negative**  | 45.42            | 6.47               | 25.99                 | 2.33                    |
>
> Table R4: Causal effects of confidence vectors.
>
> |     Steering Method   | GPQA Reflection | GPQA Backtracking | KnowLogic Reflection | KnowLogic Backtracking |
> |---------------|------------------|--------------------|-----------------------|-------------------------|
> | **Vanilla**   | 53.23            | 11.83              | 35.56                 | 5.42                    |
> | **Positive**  | 61.37            | 16.18              | 48.16                 | 5.98                    |
> | **Negative**  | 37.25            | 6.91               | 20.04                 | 3.51                    |
>
>
> [1] KnowLogic: A Benchmark for Commonsense Reasoning via Knowledge-Driven Data Synthesis

---

### Meta-Review · Area_Chair_SRvp · 2026-01-10

**Summary:**

This paper proposes RISE, an unsupervised framework that trains sparse autoencoders on sentence-level chain-of-thought activations to discover latent “reasoning vectors” corresponding to reasoning behaviors. The authors present clustering/visualization evidence that these directions organize into interpretable regions and show interventions that modulate behaviors. They further explore entropy-minimizing directions as a route to discovering “confidence”-related vectors.

The reviewers find the overall pipeline, representation construction and activation intervention to be coherent and potentially useful for mechanistic interpretability. The paper provides multiple views of structure and the rebuttal adds additional validation, such as agreement ratios. And, finally, the intervention results suggest some degree of behavior selectivity, which is a promising sign that the discovered directions are not purely artifacts of visualization.

Concerns were raised around annotation consistency, convincingness of empirical outcomes, generalization and reliance on this particular setting of LLM-as-judge.

**Reviewer Concerns:**

To complement what was written in the previous section: the rebuttal strengthens the paper when it comes to annotation consistency and cross-domain behavior steering. At the same time, the empirical outcomes still do not demonstrate clear benefits or tradeoffs on downstream objectives and it is hard to assess the practical or scientific significance of the discovered factors. Moreover, the reliance on LLM-as-Judge remains a central dependency because of remaining concerns around label validity (e.g. detailed ablations for prompt wording). The issue raised around confidence correlation with reflection/backtracking is not really resolved as separability is not clearly shown. Finally, generalization is improved in rebuttal, but overall breadth is still limited relative to the paper’s framing as “unsupervised discovery of the reasoning process.”

**Reviewer Scores:**

Given the reviews, discussion and rebuttal, all reviewers agree that the idea is novel and interesting and the paper is clearly written enough, but they are not convinced it reaches ICLR competitiveness given evidence/rigor. The rebuttal strengthens the story on annotation consistency and cross-domain generalization of steering vectors, which could have nudged two reviewers (most likely qNtY and HCkk) to increase their score by +1. However, even taking this into account, the scores remain low.

---

### Decision · Program_Chairs · 2026-01-26

Reject